# How human factors affect escalation of care: a protocol for a qualitative evidence synthesis of studies

Jody Ede,[1] Verity Westgate,[1] Tatjana Petrinic,[2] Julie Darbyshire,[1] Peter J Watkinson[1]

¹Nuffield Department of Clinical Neurosciences, University of Oxford, Oxford, UK
²Cairns Library, University of Oxford Health Care Libraries, Oxford, UK

**Correspondence to**
Jody Ede;
jody.ede@ndcn.ox.ac.uk

## ABSTRACT

**Introduction** Failure to rescue is defined as mortality after complications during hospital care. Incidence ranges 10.9%–13.3% and several national reports such as National Confidential Enquiry into Patient Outcomes and Death and National Institute of Clinical Excellence CG 50 highlight failure to rescue as a significant problem for safe patient care. To avoid failure to rescue events, there must be successful escalation of care. Studies indicate that human factors such as situational awareness, team working, communication and a culture promoting safety contribute to avoidance of failure to rescue events. Understanding human factors is essential to developing work systems that mitigate barriers and facilitate prompt escalation of care. This qualitative evidence synthesis will identify and synthesise what is known about the human factors that affect escalation of care.

**Methods and analysis** We will search MEDLINE (Ovid), EMBASE (Ovid) and CINAHL, between database inception and 2018, for studies describing human factors affecting failure to rescue and/or care escalation. A search strategy was developed by two researchers and a medical librarian. Only studies exploring in-hospital (ward) populations using qualitative data collection methods will be included. Screening will be conducted by two researchers. We are likely to undertake a thematic synthesis, using the Thomas and Harden framework. Selected studies will be assessed for quality, rigour and limitations. Two researchers will extract and thematically synthesise codes using a piloted data extraction tool to develop analytical themes.

**Ethics and dissemination** The qualitative evidence synthesis will use available published literature and no ethical approval is required. This synthesis will be limited by the quality of studies, rigour and reproducibility of study findings. Results will be published in a peer-reviewed journal, publicised at conferences and on social media.

**PROSPERO registration number** CRD42018104745.

## Strengths and limitations of this study

► Failure to rescue is a common problem in healthcare with significant effects on patient mortality.
► For failure to rescue to be avoided, an escalation of care needs to occur. The efficacy of this can be positively or negatively affected by human factors.
► This protocol ensures a comprehensive and unbiased search and analysis of qualitative studies exploring this phenomenon using best practice guidelines.
► The results of this review will identify strengths and weaknesses of the literature in this area.
► This review will highlight potential research direction for future studies and will address some of the weaknesses identified in published research projects.

(32%) reported to the National Patient Safety Agency had failures surrounding diagnostic errors and deteriorations which were not adequately recognised.[3] Failure to recognise the need to rescue patients by providing timely escalation of care is a finding in several national reports such as National Confidential Enquiry into Patient Outcomes and Death (NCEPOD)[4–7] and National Institute for Health and Care Excellence CG 50.[8]

For 'failure to rescue' to be avoided, bedside clinical staff must usually initiate successful escalation of care.[9] This staged process requires detection of deterioration, communication about deterioration and actions following senior review.[4] Many factors affect this process such as situational awareness, team working, communication, safety culture and leadership.[4 10–14] Understanding these human factors is essential to developing working systems that mitigate barriers and facilitate prompt escalation of care.

The aim of this qualitative evidence synthesis is to map the human factors which affect escalation of care in the acute hospital setting. It will summarise what is currently understood about the role human factors

## INTRODUCTION

Failure to rescue is defined as the mortality rate of patients who suffer complications in hospital.[1] The incidence of failure to rescue events varies between hospitals but has been estimated as 10.9% in high-volume hospitals and 13.3% in low-volume hospitals.[2] A proportion of severe harm and patient deaths

play in the delivery of clinical care. Second, it will identify gaps in the current literature and establish strengths and weaknesses of research conducted to date. This will produce an evidence base from which escalation of care theory could be developed. We will also identify potential areas for further research in human factors and the escalation of care process.

## METHODS AND ANALYSIS
### Registration
This protocol adheres to the requirements of Preferred Reporting Items for Systematic Reviews and Meta-Analysis Protocols (PRISMA-P).

### Information sources
Literature search strategies will be developed using Medical Subject Headings and text words related to the human factors involved in the escalation of care for deteriorating patients.

The following databases will be searched: MEDLINE (Ovid), EMBASE (Ovid) and CINAHL. Dates searched will be from database inception to January 2018.

Reference lists of eligible studies and relevant reviews will be explored to identify further eligible studies.

### Search strategy
A draft of the search strategy was developed by three of the authors (JE, VW and TP). The proposed search strategy is shown in the online supplementary file 1.

### Inclusion criteria
#### Types of studies
This qualitative evidence synthesis will include qualitative studies which report primary data. Qualitative studies are defined as those using qualitative data collection and analysis methods. These can include, but are not limited to, ethnography, interviews, focus groups and human factors methods. Data analysis is likely to include thematic analysis, grounded theory and/or discourse analysis. We will also include grey literature. All studies meeting inclusion criteria will be included and reviewed.

#### Study focus
Studies must report primary data and describe human factors affecting failure to rescue and escalation of care. Failure to rescue is defined as patient mortality following complications[1] and escalation of care is a staged process where patients are identified as 'deteriorating', and that deterioration is then communicated followed by senior review and medical intervention where necessary.[4] We will include any qualitative study which explores the perspective of patients or clinical staff (adults or paediatric) and the human factors which affect the escalation of care process. We are defining human factors as any barrier or facilitator that affects teamwork, tasks, equipment, workspace, culture or organisation.[15]

#### Setting
The study setting is in-hospital, ward care.

### Exclusion criteria
#### Types of studies
We will exclude systematic reviews, editorials, letters, practice guidelines and abstract-only reports. We will also exclude protocols without study data.

#### Phenomenon of interest
We are only interested in real-life scenarios where human factors effects can be studied in the patient environment. Simulation based studies will be excluded.

#### Setting
We will exclude studies carried out in the emergency department, critical care (including the Intensive Care Unit and coronary care) or maternity. These are specialised areas which makes it challenging to generalise to the ward environment any 'escalation of care' practices identified. We will also exclude studies set in palliative care.

#### Time frame
No time limitations will be applied.

#### Language
Non-English papers will be excluded.

### Study selection
Reference lists from all databases will be entered into Covidence software (Covidence systematic review software, Veritas Health Innovation, Melbourne, Australia. Available at www.covidence.org). Papers will be deduplicated. Two authors will independently screen titles and abstracts of identified papers against the inclusion and exclusion criteria. They will not be blinded to journal titles, study authors or institutions. If there is disagreement or uncertainty regarding eligibility, the full text will be reviewed. We will retrieve full text for all articles not excluded by the initial screening. Two authors will independently assess these papers against the inclusion and exclusion criteria outlined above. Where inclusion of a paper is uncertain, it will be fully reviewed for suitability. We will resolve disagreements about eligibility by discussion between the screening researchers or a third party. We will record the reason for excluding studies.

### Data extraction
Data extraction tools will be developed and piloted before the review takes place. Extracted data will be entered into Excel (Microsoft Office 2016). Initial coding will be documented with NVivo (NVivo qualitative data analysis Software; QSR International, V.10, 2014). Two reviewers will independently extract a selection of data from the texts to ensure validity of results. Any discrepancies within the data collection phase will be resolved by discussion between reviewers or a third party.

**Table 1** Anticipated data to be extracted

| Study Characteristics | Patient/participant demographics | Study setting | Themes | Rigour |
|---|---|---|---|---|
| ▶ Author<br>▶ Date of study<br>▶ Study type<br>▶ Methodology<br>▶ Country of study<br>▶ Data collection methods<br>▶ Journal<br>▶ Data analyses | ▶ Age<br>▶ Patient group<br>▶ In-patient characterisation | ▶ Level of care<br>▶ Hospital type<br>▶ Education | ▶ Codes | ▶ Strengths<br>▶ Weaknesses<br>▶ Reporting guidelines used |

### Data items extracted

We will extract the following data from each included publication (refer to table 1 for full data details). The data extraction method has been piloted with a sample selection of papers and valid data have been obtained.

### Quality assessment

The Critical Appraisal Skills Programme (CASP) qualitative checklist will be used to assess credibility, transferability, dependability and confirmability. This checklist is an extensive and comprehensive tool commonly used in qualitative study assessment.[16 17] As part of the CASP assessment the authors will explore the potential for reporting bias within the studies and biases will be reported in studies' limitations. Two researchers will discuss each study and a consensus will be reached to include or exclude.

### Assessment of confidence in synthesised findings

We will apply the Grading of Recommendations Assessment, Development, and Evaluation, Confidence in the Evidence from Reviews of Qualitative research (GRADE-CERQual) criteria to judge confidence in synthesised findings.[18] We will apply the CERQual criteria to each study finding, assessing for methodological limitations, relevance, coherence and adequacy of data. This method will generate a summary of qualitative study findings table, providing a transparent method with which to assess included studies and results.[18]

### Data analysis

This review aims to explore relevant theory and map barriers and facilitators to escalation of care for which thematic synthesis is well suited.[17] We are likely to undertake a thematic synthesis, using the Thomas and Harden framework.[19] This framework supports data extraction from anywhere within the paper, and is not confined to the results alone. The three stages of the framework are: coding findings from included studies, categorisation of codes into descriptive themes and categorisation of descriptive themes into analytical themes.[19] Stage one involves line by line coding of data, where each sentence is allocated a code. Stage two involves categorising each coded sentence into descriptive, broader themes. The final stage involves generating analytical themes, or 'going beyond' the findings of the initial study, which relate to the fixed or emerging research question. While we have been explicit at this point as to the anticipated framework, it is also justifiable for this to change once the search has been conducted.[20]

NVivo software will be used to code the original text from papers. Using this software will facilitate analysis for this evidence synthesis and will be used to record decisions (by audit trail) of coding. Codes relating to human factors and escalation of care will be identified from anywhere within the papers, and tables will be used to record descriptive and analytical themes. Key codes, descriptive themes and analytical themes will be presented in the results. We will use the enhancing transparency in reporting the synthesis of qualitative research guidelines to report findings.[21]

### Patient and public involvement

A patient representative (TD) has read and provided feedback on the protocol. As a result, some points have been clarified and medical 'jargon' removed.

### ETHICS AND DISSEMINATION

The proposed evidence synthesis will use published literature and therefore no ethical approval is required. This publication will be limited by the quality of studies available and the rigour and reproducibility of study findings. Original studies included in the review could themselves be limited and it may be difficult to assess the researcher involvement and their individual bias. The two researchers carrying out screening for this review come from different professional backgrounds, limiting interpretation bias when assessing the studies for inclusion. A recognised assessment tool will be used to determine study quality. Using NVivo to code studies will aid transparency and demonstrate a clear strategy for theme identification. An audit trail kept throughout the process, will detail decisions made and methodological steps taken.

The results from this review will be published and made freely available. A number of social media techniques (including Twitter, Facebook, and our research group

website) will be used to promote the protocol, final paper and results. We will also aim to attend at least one conference to present findings from this work.

**Acknowledgements** We would like to thank patient representative, TD for his contribution to this work.

**Contributors** PW is the guarantor. JE was responsible for the overall design of the QES. JE and VW drafted the manuscript. TP and JE developed the search strategy. PW and JD provided QES and qualitative expertise. All authors read, provided feedback and approved the final manuscript.

**Funding** This QES protocol is funded by NIHR Biomedical Research Centre, based at Oxford University Hospitals Trust, Oxford and the Department of Health and Wellcome Trust through the Health Innovation Challenge Fund. This publication presents independent research commissioned by the Health Innovation Challenge Fund (HICF-R9-524; WT-103703/Z/14/Z), a parallel funding partnership between the Department of Health and Wellcome Trust. This work was also supported by the NIHR Biomedical Research Centre, Oxford.

**Disclaimer** The views expressed are those of the author(s) and not necessarily those of the NHS, the NIHR or the Department of Health.

**Competing interests** None declared.

**Patient consent for publication** Not required.

**Provenance and peer review** Not commissioned; externally peer reviewed.

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
