## [Reviewer comments · BMJ Open]

ARTICLE DETAILS

TITLE (PROVISIONAL)	How human factors affect escalation of care: a protocol for a qualitative evidence synthesis of studies
AUTHORS	Ede, Jody; Westgate, Verity; Petrinic, Tatjana; Darbyshire, Julie; Watkinson, Peter

VERSION 1 – REVIEW

REVIEWER	Jane Noyes Bangor University, UK
REVIEW RETURNED	12-Sep-2018

GENERAL COMMENTS	Thank you for asking me to review this protocol for a qualitative evidence synthesis. I have two main areas of concern: 1. The manuscript would benefit from the input of lay advisers (ie patients and public representatives) to translate the somewhat clunky text with a lot of medical jargon into plain English.2. The methods and selection of Thomas and Harden's approach to thematic synthesis requires greater articulation and justification. This 3 stage approach is designed to develop theory and transcend findings in individual primary studies to develop new insights that were not apparent in single studies. The aim and questions as currently stated do not elude to this or provide any clue as to why this method has been selected over and above other methods (that serve different purposes). Thomas and Harden's approach usually entails coding onto the entire manuscript using a software product (this is how Thomas and Harden operationalised their approach so as not to lose the original context), rather than data extraction approach as described in the protocol. Thomas and Harden are clear that findings can be located in the entire paper and not just the findings section in a qualitative primary study report. The description of the 3 stages of synthesis is thin and there is no mention of engagement with consumers and key stakeholders to interpret the evidence and build new theory. The language used is a bit of a hybrid (primary outcomes and phenomenon of interest) and is not entirely consistent with a qualitative evidence synthesis. For example a QES does not usually have primary and secondary outcomes. Suggest stick to aims and exploring phenomenon of interest and ditch primary and secondary outcomes that are more aligned with the terminology used in a quant SR. A QES question formulation framework such as SPICE may help with greater alignment of the review question and purpose with the selected methods. The authors may want to consider including studies reported as grey literature - especially given the topic and context. There is no mention as to how the outcome of quality assessment appraisals will be used in the review (for example how will methodological limitations in primary
---

	studies be used when developing and interpreting findings?). The authors may want to consider using GRADE CERQual to assess the confidence in synthesised qualitative findings. There is no indication as to whether all studies that meet the inclusion criteria will be included or whether the authors would consider sampling and if so based on what theoretical criteria. As a final caveat, many QES authors opt not to select their method of synthesis until they are aware of the amount, type and quality of the primary studies for synthesis. Conceptually rich studies (within the sample) are preferable for a theory development method such as Thomas and Harden's approach - if the studies are conceptually thin (which many clinical study qualitative report are) then Framework synthesis or Best fit Framework synthesis may be the method of choice. The INTEGRATE guidance outlines criteria for selecting a method in these circumstances. https://www.integrate-hta.eu/wp-content/uploads/2016/02/Guidance-on-choosing-qualitative-evidence-synthesis-methods-for-use-in-HTA-of-complex-interventions.pdf. Authors now commonly build in some flexibilities to their QES protocols in case the preferred method turns out not to be the ideal method once the pool of evidence is known. I could not find mention (but may have missed it) of the ENTREQ reporting guidelines for QES reviews. Nor could I find mention of consumer and key stakeholder engagement/input in protocol development and subsequent review processes. Overall, and clinically important and interesting topic. The QES protocol is thin on methodological detail and the language sometimes veers off into the quantitative SR domain. All these issues are fixable with some rewriting and I would urge the authors to do this. Best of luck with conducting your QES.
--	--

REVIEWER	Amir Ghaferi University of Michigan; USA
REVIEW RETURNED	25-Sep-2018

GENERAL COMMENTS	No clarification needed in the study protocol. No major flaws identified. I look forward to reading the study results.
--

REVIEWER	Professor Ruth Endacott University of Plymouth, UK, and Monash University, Australia
REVIEW RETURNED	30-Oct-2018

GENERAL COMMENTS	Clear protocol for the review; methods are appropriate and sound rationale provided. It might be wise to add CINAHL database as some studies might otherwise be missed.ensure all acronyms are explained in full at first use (e.g. in the main text Introduction, FTR and NICE CG). Acronyms should be avoided in the abstract. in the Introduction section amend text to: "several national reports published by the National...".
--

VERSION 1 – AUTHOR RESPONSE

Thank you very much to you and your reviewers, whose comments we have found most useful. Comments are addressed individually below:

Editor comments:

C- Please include the dates of the search in both the abstract and the main methods sections.
R-Searches are yet to be conducted.

C- Please ensure that the arrangements of authors in your main document and Scholar One submission system are the same.
R-Amended

C-The in-text citation for 'Table 1' is missing in your main text of your main document file. Please amend accordingly.
R-Amended (Page 5, line 8)

C -The manuscript would benefit from the input of lay advisers (i.e. patients and public representatives) to translate the somewhat clunky text with a lot of medical jargon into plain English.
R-A patient representative has read this protocol and I have made some further amendments to reduce the medical jargon.

C-The methods and selection of Thomas and Harden's approach to thematic synthesis requires greater articulation and justification
R-We have taken this reviewer's extensive comments on board and have justified the use of the Thomas and Harden framework in more detail. We have also left some flexibility in the protocol to change this framework once the data has been reviewed. I found the link to the PDF document one of the most helpful resources that I have encountered during this process. (Page 5, line 22-25)

C-The description of the 3 stages of synthesis is thin
R-The three stages of synthesis are now outlined in more detail (Page 5, line 15-31).

C-The language used is a bit of a hybrid (primary outcomes and phenomenon of interest) and is not entirely consistent with a qualitative evidence synthesis
R-We have revised the structure of the paper to avoid describing primary outcomes (Page 3, line 15-20).

C-There is no indication as to whether all studies that meet the inclusion criteria will be included or whether the authors would consider sampling and if so based on what theoretical criteria
R-All studies meeting the inclusion criteria will be included in the synthesis (Page 4, line 6-7).

C-The authors may want to consider using GRADE CERQual to assess the confidence in synthesised qualitative findings
R-We have included this as one of the assessment criteria (Page 5, line 18-19).

3rd Reviewer's comments:

C-It might be wise to add CINAHL
R- The CINAHL database has been added as a source of publications (Page 2, line 14 and Page 3, Line 30).

C-Acronyms should be avoided in the abstract.
R-These have been removed.

C-No discussion about PPI
R- A sentence about PPI done for this evidence synthesis has been added. (Page 6, lines 8-10)

VERSION 2 – REVIEW

REVIEWER	Jane Noyes Bangor University, UK
REVIEW RETURNED	16-Jan-2019

GENERAL COMMENTS	A nicely done revision that has further strengthened the manuscript. Two outstanding issues: 1. The usual convention for systematic reviews is to unpick them and screen potentially relevant studies for inclusion.2. The authors are confused about the application of GRADE CERQual. They wrongly assert that CERQual is applied at individual study level. It is applied at the level of synthesised findings. This needs correcting. To avoid confusion the authors should consider inserting another subheading ' Assessment of confidence in synthesised findings' and describe the application of CERQual under this new heading.
--

VERSION 2 – AUTHOR RESPONSE

C-Please include the dates of the search: this refers to the time interval being searched in the databases, e.g. publications from January 2010 to December 2018

A- I have added search dates (The databases were searched from inception to January 2018) Page 3, Line 30

C-The usual convention for systematic reviews is to unpick them and screen potentially relevant studies for inclusion

A-Page 4, Line 43

C-The authors are confused about the application of GRADE CERQual. They wrongly assert that CERQual is applied at individual study level. It is applied at the level of synthesised findings. This needs correcting. To avoid confusion the authors should consider inserting another subheading ' Assessment of confidence in synthesised findings' and describe the application of CERQual under this new heading.

A- Please see amended text under sub-heading of 'Assessment of confidence in synthesised findings' Page 5, 24-28